# A Novel Recurrent 200 kb *CRYL1* Deletion Underlies DFNB1A Hearing Loss in Patients from Northwestern Spain

**DOI:** 10.3390/genes16060670

**Published:** 2025-05-30

**Authors:** Guadalupe A. Cifuentes, Marta Diñeiro, Alicia R. Huete, Raquel Capín, Adrián Santiago, Alberto A. R. Vargas, Dido Carrero, Esther López Martínez, Beatriz Aguiar, Anja Fischer, Roland Rad, María Costales, Rubén Cabanillas, Juan Cadiñanos

**Affiliations:** 1Fundación Centro Médico de Asturias, 33193 Oviedo, Spain; 2Instituto de Medicina Oncológica y Molecular de Asturias (IMOMA), 33193 Oviedo, Spain; 3Hospital Centro Médico de Asturias, 33193 Oviedo, Spain; 4Institute of Molecular Oncology and Functional Genomics, School of Medicine, Technische Universität München, 81675 Munich, Germany; 5Hospital Universitario Central de Asturias, 33011 Oviedo, Spain; 6Cabanillas Precision Consulting, 8620 Zürich, Switzerland

**Keywords:** hereditary hearing loss, genetic diagnostics, DFNB1A, *GJB2*, *CRYL1*, large deletions, cis-regulation

## Abstract

Background/Objectives: Pathogenic recessive *GJB2* variants are the main genetic cause of non-syndromic sensorineural hearing loss. However, following *GJB2* testing, a significant proportion of deaf patients are only found to be heterozygous carriers of pathogenic *GJB2* alleles. Five large deletions not affecting *GJB2* but encompassing a minimal common 62 kb region within the neighbouring *CRYL1* gene have been described to cause loss of *cis GJB2* expression and, as a result, produce hearing loss when in *trans* with pathogenic *GJB2* variants. We describe the identification and characterization of a novel deletion of this type in deaf patients from northwestern Spain. Methods: We used panel NGS sequencing to detect the deletion, MLPA to validate it, whole-genome sequencing to map its breakpoints, PCR + Sanger sequencing to finely characterize it and triple-primer PCR to screen for it. Results: We identified a novel 200 kb deletion spanning the whole *CRYL1* gene in two unrelated deaf patients from Asturias (in northwestern Spain) who were heterozygous for the pathogenic *GJB2* c.35delG variant. Although the large deletion was absent from gnomAD v4.1.0 and 2052 local control alleles, screening for it in 20 additional deaf carriers of monoallelic pathogenic *GJB2* variants detected it in another patient from Galicia (also in northwestern Spain). The novel deletion, termed del(200 kb)insATTATA, explained hearing loss in 3/43 (7%) deaf patients from our cohort that were otherwise heterozygous for pathogenic *GJB2* variants. Conclusions: This work highlights the importance of comprehensively testing all genomic regions known to be clinically relevant for a given genetic condition, including thorough *CRYL1* CNV screening for DFNB1A diagnostics.

## 1. Introduction

Hearing loss (HL) is the most common sensory disorder, affecting approximately 5% of the world population [1]. Sensorineural hearing loss (SNHL), the condition with the highest prevalence, is caused by damage to the cochlea, the auditory nerve or the brain’s central processing centres. Although most deaf patients are adults, HL is present congenitally in 1–2 out of 1000 newborns [2]. Genetic causes account for 50–60% of childhood HL cases in developed countries, and 70% of these correspond to non-syndromic SNHL (NS-SNHL). With regard to inheritance patterns, 75–80% of such cases are autosomal recessive (AR), while 20% are autosomal dominant (AD), close to 2% are X-linked and less than 1% are mitochondrial [3].

In spite of the high genetic heterogeneity of SNHL, with hundreds of genes involved, pathogenic variants in the *GJB2* gene alone, encoding connexin 26, are responsible for up to half of severe to profound NS-SNHL in many western populations [4]. To date, more than 100 pathogenic *GJB2* variants resulting in HL have been identified. The majority of these are loss-of-function variants associated with the autosomal recessive 1A deafness (DFNB1A) phenotype (MIM #220290). In Caucasians, *GJB2* (NM_004004.6) c.35delG is the most frequent DFNB1A pathogenic variant, comprising 70% of pathogenic DFNB1A alleles in some cohorts, and with a carrier rate of 1–3% in the general population [5].

Apart from small (single nucleotide and indels) and large (copy-number) loss-of-function variants directly affecting the *GJB2* locus, to date, five different large deletions not spanning the *GJB2* locus have been described to cause hearing loss when they occur in compound heterozygosity with another pathogenic DFNB1A variant (Table 1 and Figure 1A) [6,7,8,9,10]. The first two such deletions identified, del(*GJB6*-D13S1830) (spanning 309 kb) and del(*GJB6*-D13S1854) (spanning 232 kb), affected the *GJB6* locus, immediately telomeric to *GJB2*, leading to the assumption that *GJB6* was also involved in non-syndromic hearing loss in a digenic fashion with *GJB2* [6,7]. However, subsequent functional studies indicated that the hearing loss associated with the large *GJB6* deletions was not caused by a loss of function in *GJB6*, but by loss of *GJB2* expression [11,12,13]. In parallel, two novel large neighbouring deletions (spanning 131 kb and 179 kb, respectively) were identified in deaf patients with monoallelic pathogenic *GJB2* variants [8,9]. These novel deletions did not affect the *GJB6* locus. Instead, they overlapped with the telomeric halves of del(*GJB6*-D13S1830) and del(*GJB6*-D13S1854), defining a 95.4 kb region shared by the four of them and affecting the *CRYL1* locus. Finally, recent work studying hearing loss in an East-Asian cohort (all previous *CRYL1* deletion had been originally identified in individuals with European ancestries), detected a recurrent 125 kb *CRYL1* deletion shared by several unrelated deaf patients with monoallelic pathogenic *GJB2* variants [10]. The minimal common region (MCR) shared by all these deletions is 62 kb long and spans the 3′ half of *CRYL1* (which is in the minus strand as *GJB2* and *GJB6*) plus 5.3 kb further downstream [chr13(GRCh38):20398370-20460629] (Figure 1A). Such MCR includes a candidate 1043 bp regulatory region termed C3 [chr13(GRCh38):20419404-20420446], containing potential enhancers that have been functionally demonstrated to affect *GJB2* expression [14].

In this work, combining the OTOgenics^TM^ NGS panel [15], Whole Genome Sequencing (WGS) and breakpoint-specific PCR, we have identified and genomically characterized a novel 200 kb *CRYL1* deletion present in heterozygosis in three unrelated deaf patients from northwestern Spain who were also heterozygotes for the frequent *GJB2* (NM_004004.6) c.35delG pathogenic variant. The novel deletion includes the 62 kb MCR, is not present in individuals from gnomADv4 nor in 2052 control local chromosomes and segregates with hearing loss when in compound heterozygosity with *GJB2* (NM_004004.6) c.35delG.

## 2. Materials and Methods

### 2.1. Patients

We identified the novel deletion in two patients (Patients #1 and #2) with hearing loss who underwent NGS with a gene panel comprising > 230 genes associated with sensorineural or mixed, syndromic or non-syndromic, HL [15]. The deleted region included 10 probes distributed across the *CRYL1* locus (in the minus strand), plus 1 probe upstream and 5 probes downstream of it (affected probes shown in red in Figure 1A). Then, we tested for the presence of the novel *CRYL1* deletion in deaf patients (either wild-type for *GJB2* or monoallelic for NM_004004.6: c.35delG) with no genetic diagnosis after previous analyses based on Sanger sequencing or the NGS panel. Samples from normal hearing individuals were taken from our lab research sample collections. All involved individuals had provided written informed consent.

### 2.2. Novel Deletion Detection, Validation and Characterization

The novel large *CRYL1* deletion was detected in our laboratory using the previously described OTOgenics^TM^ panel [15]. Orthogonal validation was initially performed through MLPA on Patient #1’s and Patient #2’s germline DNA, using the P163-E1 GJB-WFS1-POU3F4 kit and following the manufacturer’s instructions (MRC Holland, Netherlands). The *CRYL1* deletion breakpoint regions were identified through WGS of Patient 2’s germline DNA at Macrogen (Macrogen Inc., Seoul, South Korea), and they were then finely mapped by PCR using two primers flanking the breakpoint regions (5′-TTTCCTCATCCAACTGCCCA-3′ and 5′-TTCGAAGGTACAGGGGAGAC-3′), followed by Sanger sequencing. That PCR, performed using MegaMix-Double reagent (Microzone, Stourbridge, UK) under the manufacturer’s suggested conditions, generating an 897 bp fragment in the presence of the deletion-containing allele.

### 2.3. del(GJB6-D13S1854), del(GJB6-D13S18530) and CRYL1 del(200 kb)insATTATA Deletion Screening

del(*GJB6*-D13S1854) and del(*GJB6*-D13S1830) were screened for as previously described [6,7]. For evaluating the presence of the novel *CRYL1* del(200 kb)insATTATA, we designed a triple-primer PCR that generates two differential amplicons depending on the allele, which are easily discerned in an electrophoresis gel. We used two primers flanking the deletion breakpoint (5′-GATTGTGGAATGCCCAAAGT-3′ and 5′-TGCTCAATTGACACCAACAA-3′) that amplify a 436 bp-product from genomic DNA bearing the deletion (including the 6 bp from the ATTATA insertion) and a 200 kb fragment from the wild-type allele (not amplifiable in normal PCR conditions); the third primer (5′-TTTACTGAGGCTGCCGGTAT-3′) anneals within in the deleted region and amplifies, together with the first primer, a 702 bp product that includes sequence from the deleted interval. For the PCR we used MegaMix-Double reagent (Microzone, Stourbridge, UK) following the manufacturer’s suggested conditions, with an annealing temperature of 63 °C, and we visualized the products after electrophoresis in an 1.2% agarose gel.

### 2.4. GJB2 c.35delG Genotyping

To determine the presence/absence of *GJB2* (NM_004004.6) c.35delG in the two brothers of Patient #2, we used Megamix Double (Microzone, UK) and chimeric primers with sequences complementary to M13 and human *GJB2* (5′-tgtaaaacgacggccagtCCCAGCACAGCAAATTTTTA-3′ and 5′-tgtaaaacgacggccagtCCCAGCACAGCAAATTTTTA-3′; lowercase: oligonucleotide part specific M13D or M13R sequencing primers, respectively; uppercase: oligonucleotide part specific for human *GJB2*) to amplify a 723 bp amplicon, that we Sanger-sequenced with M13D and M13R primers at the Scientific and Technical Services of University of Oviedo (Spain).

## 3. Results

### 3.1. Detection of a Novel CRYL1 Deletion in Hearing Loss Patients Using an NGS Panel

With the aim of identifying the genetic cause of deafness in Patient #1, diagnosed with profound sensorineural hearing loss at the age of 5 months, we used the OTOgenics^TM^ NGS panel (v4) to sequence 231 genes associated with hearing loss in genomic DNA from peripheral blood cells. The panel contains probes to capture exons and exon-intron boundaries from all target genes, as well as a selection of additional probes to capture genomic regions bearing previously described non-coding pathogenic variants. The latter include 27 probes (281–505 nt) scattered from the intergenic region upstream of *GJB6* to the intergenic region upstream of *CRYL1* intron (both genes lying on the minus strand of chromosome 13), with an average separation of 10,888 (6120–22,320) bp between them (Figure 1A). Sequence analysis detected a pathogenic *GJB2* (NM_004004.6) c.35delG variant in heterozygosis in Patient #1 as the only clinically relevant variant directly affecting *GJB2*. However, copy-number analysis of NGS results detected a potential 50% loss in the relative read depth of all 10 probes targeting *CRYL1* intronic regions as well as the first 5 intergenic probes placed between *CRYL1* and *GJB6* (Figure 1B). The presence of a heterozygous deletion affecting *CRYL1* but not *GJB6* was confirmed by MLPA (Figure 1C). The pattern of probes showing decreased read depth indicates that this is a novel *CRYL1* deletion, not matching any of the large deletions previously described in this genomic region (Figure 1A,B). As this variant encompassed the minimal common region shared by all previously reported pathogenic *CRYL1* variants, and taking into account the published evidence on large deletions contained within the probes that detected 50% read depth loss in our patient, we classified it as pathogenic according to ACMG guidelines (PM3_VeryStrong, PP1_Strong, PM2_Supporting) [8,10,16,17], and issued a clinical report considering it as causative of DFNB1A in the patient in conjunction with *GJB2* (NM_004004.6) c.35delG, provided that the variants affected different alleles (*trans* configuration).

A few months later, we found an identical copy-number pattern from the OTOgenics^TM^ (v5) NGS results of a new deaf patient (Patient #2). Patient #2, unrelated to Patient #1, was diagnosed with profound sensorineural hearing loss at the age of 2 years. He also had visual impairment and intellectual disability, which may have been the result of complications during childbirth. Patient #2 was heterozygous for *GJB2* (NM_004004.6) c.35delG and his gDNA showed the same regional read depth (and MLPA) profiles as Patient #1’s (Figure 1B,C). We hypothesized then that we might have found a local/regional *CRYL1* deletion that could be responsible for other unexplained hearing loss cases in our area (northwestern Spain), so we set out to genomically characterize it with the aim of designing a quick and cost-effective PCR assay to screen for it in other deaf patients with heterozygous pathogenic/likely pathogenic *GJB2* variants.

### 3.2. Mapping of the Breakpoints of the New CRYL1 Deletion

The pattern for relative read depth on the probes situated around *GJB6* and *CRYL1* suggested that the centromeric breakpoint of the deletion was placed within the chr13(GRCh38/hg38): 20352212-20362132 interval. However, it did not provide information to infer a candidate region for the telomeric end, as the most telomeric probe also showed read depth loss (Figure 1A,B). Thus, we performed WGS on gDNA from Patient #2 and focused on the genomic area under study, revealing a clear reduction in coverage depth and absence of heterozygosity that spanned approximately chr13(GRCh38/hg38):20360000-20560000 (Figure 2A). Closer examination of the boundaries of such region allowed detection of both discordant-read pairs and split reads supporting the existence of a centromeric breakpoint at chr13(GRCh38/hg38):20361159-20361160 and a telomeric breakpoint at chr13(GRCh38/hg38):20561391-20561392 (Figure 2B).

We designed PCR primers (chr13(GRCh38/hg38):20360608-20360627, plus strand; chr13(GRCh38/hg38):20561711-20561730, minus strand) surrounding the candidate breakpoints to amplify a deletion-specific amplicon from both Patient #2 and Patient #1 (Figure 1A, Figure 2B and Figure 3A). Sanger sequencing of the amplicon confirmed the breakpoints at chr13(GRCh38/hg38):20361159-20361160 and chr13(GRCh38/hg38): 20561391-20561392 and revealed a 6-base (ATTATA) insertion (Figure 3 A,B). Thus, HGVS nomenclature of the novel deletion is chr13(GRCh38/hg38):20361160_20561391delinsATTATA. A search in the structural variant dataset of the gnomAD v4.1.0 database, able to identify all other large deletions represented in Figure 1A except for del(179 kb), did not detect the novel *CRYL1* del(200 kb)insATTATA variant, indicating that this is a previously undescribed alteration.

### 3.3. Novel Deletion Screen in Unexplained Hearing Loss Cases and in the Normal Hearing Population

As Patient #1 and Patient #2 were not family-related, we hypothesized that the novel deletion might also be the second hit responsible for hearing loss in other deaf patients. Thus, to screen for it, we designed a novel pair of primers closer to the breakpoints [chr13(GRCh38/hg38):20360924-20360943, plus strand; chr13(GRCh38/hg38):20561566-20561585, minus strand]. Introduction of a second reverse primer (chr13[GRCh38/hg38]:20361606-20361625, minus strand) annealing 3′ to the 5′ breakpoint allowed us to design a triple-primer PCR expected to produce a 702 bp amplicon for the wt allele and a 436 bp amplicon (including the 6 bases from the ATTATA insertion) from the deleted allele (Figure 1A, Figure 2B and Figure 3C). We used this triple-primer PCR assay to analyze gDNA from 20 patients previously evaluated in our laboratory who had not been tested for *CRYL1* deletions and who were heterozygous for a pathogenic/likely pathogenic *GJB2* variant and negative for the previously described del(*GJB6*-D13S1830) and del(*GJB6*-D13S1854) large deletions. As a result, we obtained the expected amplicon from another *GJB2* c.35delG heterozygous patient (Patient #3, from Galicia) who had been referred to our laboratory for genetic testing due to childhood hearing loss (Figure 3C). Finally, we did the same triple-primer PCR on DNA samples from 14 deaf patients previously analyzed in our laboratory with versions of the OTOgenics panel that contained probes against the *CRYL1* region, that were found to be heterozygous for pathogenic recessive *GJB2* variants, negative for del(*GJB6*-D13S1830) and del(*GJB6*-D13S1854) and in whom *CRYL1* deletions had not been detected through the NGS assay, obtaining (as expected) only the wild-type band (Appendix A). Genetic analysis of Patient #2’s siblings showed that del(200 kb)insATTATA segregates with hearing loss when in compound heterozygosity with *GJB2* (NM_004004.6) c.35delG (Figure 3D), adding more evidence in favour of the PM3 ACMG criterion previously considered for its classification as a pathogenic variant [8,10,16,17].

We then used this assay to test gDNAs from 285 other deaf patients previously evaluated for suspected hereditary hearing loss in our laboratory, for whom a genetic cause had not being identified, and who were not known to be carriers of pathogenic DFNB1A variants. We found that all of them were negative for the novel deletion. After that, to explore the regional population frequency of chr13(GRCh38/hg38): 20361160_20561391delinsATTATA, we analyzed saliva gDNA from 1026 local individuals (2052 alleles) without reported suspected hereditary hearing loss, none of which was a carrier of the novel variant (allele frequency < 0.00049). Thus, we have identified a novel, large, rare *CRYL1* deletion, found in heterozygosis in three unrelated deaf patients from northwestern Spain (2 from Asturias and 1 from Galicia) who are also heterozygous for *GJB2* (NM_004004.6) c.35delG.

### 3.4. Diagnostic Relevance of CRYL1 del(200 kb)insATTATA Within Deaf Patients with Monoallelic Pathogenic GJB2 Variants

To estimate the potential diagnostic impact of *CRYL1* del(200 kb)insATTATA in our population, we determined its prevalence within deaf patients with monoallelic pathogenic *GJB2* variants that had followed different diagnostic paths, and compared it with the prevalence of del(*GJB6*-D13S1830), of del(*GJB6*-D13S1854) and of causative variants in non-DFNB1A genes.

In total, we considered 43 deaf patients with monoallelic pathogenic *GJB2* variants. In 13 of them, no NGS panel testing had been requested (Figure 4A). Of those 13, heterozygous del(*GJB6*-D13S1830) was found in three (23.1%), whereas heterozygous del(200 kb)insATTATA was detected in one (7.7%) (Patient #3). NGS panel testing had been requested for the other 30 (Figure 4B). Of those, four patients (13.3%) were found to have heterozygous *GJB6* deletions [one del(*GJB6*-D13S1830) (3.3%) and three del(*GJB6*-D13S1854) (10%)], and two patients (6.6%) were found to have heterozygous del(200 kb)insATTATA (Patients #1 and #2). Causative variants in non-DFNB1A genes were found in 8 of the 30 patients with monoallelic pathogenic *GJB2* variants than underwent panel testing (26.6%) (Figure 4B).

Thus, within 43 deaf carriers of monoallelic pathogenic *GJB2* variants, heterozygous large deletions affecting *CRYL1* explained hearing loss in 10/43 (23.3%): 4/43 (9.3%) being del(*GJB6*-D13S1830, 3/43 (7%) being del(*GJB6*-D13S1854) and 3/43 (7%) being del(200 kb)insATTATA (Figure 4C). These results show that deletions affecting *CRYL1* different from del(*GJB6*-D13S1830) and del(*GJB6*-D13S1854) have a relevant impact in DFNB1A diagnosis and, thus, should not be overlooked, but actively screened for, particularly in deaf carriers of monoallelic pathogenic *GJB2* variants.

## 4. Discussion

Genetic alterations causing *GJB2* dysfunction are the most prevalent cause of hereditary hearing loss worldwide. These include not only genetic variants directly affecting the *GJB2* locus, but also distant rearrangements that eliminate *cis*-acting regulatory elements necessary for *GJB2* expression. In this work, we identified a novel 200 kb deletion spanning more than 20 kb each side of the *CRYL1* locus and present in heterozygosis in three unrelated hearing loss patients from northwestern Spain who are also heterozygous for the most prevalent pathogenic *GJB2* mutation in Europeans (c.35delG). This novel *CRYL1* deletion has not been detected in individuals from gnomADv4.1.0, nor in 2052 alleles from a control local population, and its presence in compound heterozygosity with c.35delG segregates with hearing loss.

This study, together with previous works describing other large deletions involving *CRYL1* (Figure 1) in DFNB1A patients, demonstrates that genetic testing of the *GJB2* locus, even if complemented with approaches targeting the known *GJB6* and *CRYL1* deletions already described to alter *GJB2* expression, may leave a significant proportion of DFNB1A patients undiagnosed. Thus, del(200 kb)insATTATA explained 3 out of 43 (7%) DFNB1A cases, in which, otherwise, only a monoallelic pathogenic/likely pathogenic *GJB2* variant had been detected in our laboratory. This is the same number of cases as those explained by the known del(*GJB6*-D13S1854) deletion in this very cohort (Figure 4C). In our opinion, all genetic diagnostic strategies for hereditary deafness that claim to be comprehensive should be able to detect any copy number alteration affecting the 62 kb *CRYL1* MCR. As illustrated in Figure 1, MLPA might not be sufficient for this, as only one of the two probes included in the commercial kit used in this work overlaps with the MCR. Moreover, this probe is close to its telomeric end and more than 10 kb away from the C3 region.

In line with EMQN Best Practice recommendations for population-tailored initial *GJB2* testing [5] and leveraging our previously validated OTOgenics™ NGS panel [15], we advocate for a two-step DFNB1A workflow. The first step should be *GJB2* sequencing plus detection of known large deletions spanning the *CRYL1* MCR. Unsolved cases after the first step would be analyzed by a comprehensive NGS panel able to detect all relevant *CRYL1* deletions, to maximize diagnostic yield of hearing loss in clinical settings.

In conclusion, in this work, we have identified a novel large *CRYL1* deletion that, in combination with heterozygous *GJB2* c.35delG, causes DFNB1A hearing loss in deaf patients from northwestern Spain, and we describe a simple triple-primer PCR assay that could be used to detect the novel large deletion. Considering the historically relevant migratory patterns of people from Asturias and Galicia to other regions of Spain, Europe and Latin America, this variant might be responsible for DFNB1A cases in several geographically distant populations.

Consequently, we invite clinical genetics laboratories with sample collections from deaf patients of possible Spanish ancestry without a genetic diagnosis who are carriers of monoallelic DFNB1A pathogenic/likely pathogenic variants to screen them for the presence of the *CRYL1* del(200 kb)insATTATA deletion described herein.

## Figures and Tables

**Figure 1 genes-16-00670-f001:**
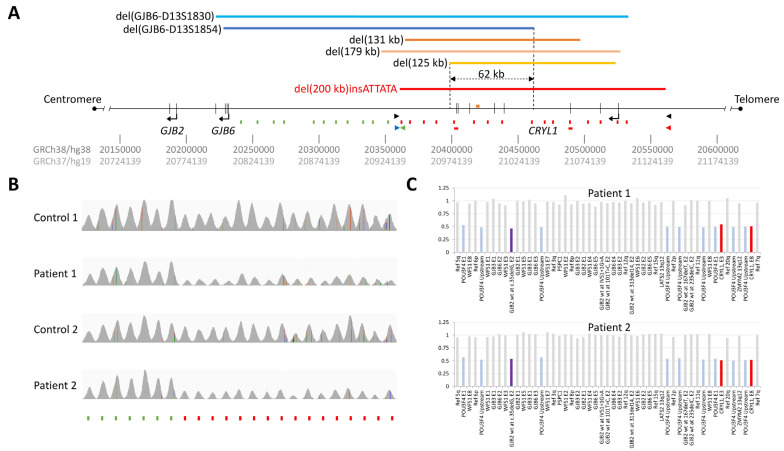
Identification of the novel 200 kb *CRYL1* deletion. (**A**). Schematic representation of chromosome 13 genomic regions affected by DFNB1A-causative large deletions that involve *CRYL1* (+/− *GJB6*) but not *GJB2.* Light-blue [6], dark-blue [7], orange [8], beige [9] and yellow [10] segments correspond to previously described deletions, whereas the novel 200 kb *CRYL1* deletion described in this work is shown in red. The double headed dotted arrow represents the 62 kb overlapping segment. Short vertical green/red segments indicate the positions of the probes included in our capture design to detect large DFNB1A deletions (red ones: affected by the novel deletion; green ones: unaffected), the short orange horizontal segment indicates the C3 region and the short red horizontal segments signal the position of the two *CRYL1* probes (targeting exon 8 and exon 3) contained in the MLPA kit used for validation. Black arrowheads mark the positions and orientations of the primers used in the first PCR designed to amplify the breakpoint region, whereas blue, green and red arrowheads represent the oligonucleotides used in the triple-primer PCR screening assay. GRCh38/hg38 and GRCh37/hg19 genomic coordinate scales are shown at the bottom. Note that *GJB2*, *GJB6* and *CRYL1* are all in the minus strand. (**B**). NGS read depths (as seen on IGV) obtained by OTOgenics panel sequencing of germline DNAs from Patient #1 and Patient #2 on the deleted region (captured by the probes represented at the bottom by red segments) and the adjacent non-deleted region centromeric to *CRYL1* (captured by the probes represented at the bottom by green segments), each compared to those from a control individual (not affected by the *CRYL1* deletion) sequenced in the same run and with a similar total number of reads. (**C**). MLPA analysis of Patient #1 and Patient #2 germline DNA. Red bars correspond to probes targeted to *CRYL1* exon 3 and exon 8 (showing heterozygous deletion in both patients). The purple bar corresponds to a probe targeting the wild-type region affected by the *GJB2* c.35delG pathogenic variant (heterozygous in both patients). Blue bars correspond to probes targeting the POU3F4 gene, located in the X chromosome (both Patient #1 and Patient #2 are males).

**Figure 2 genes-16-00670-f002:**
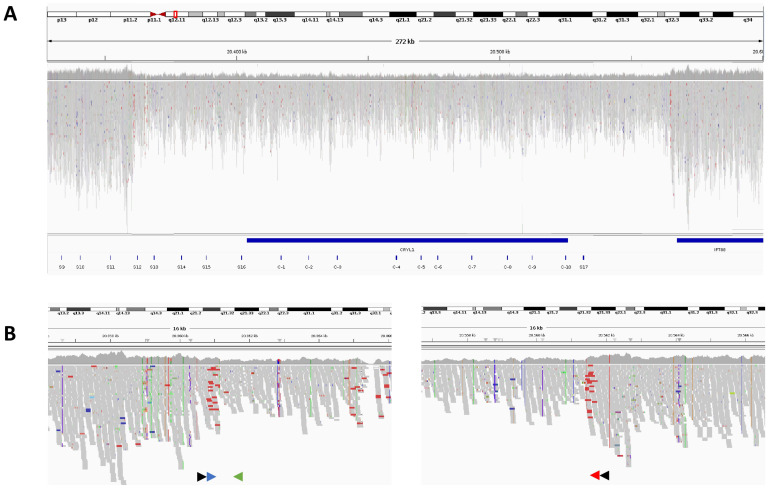
Visual analysis of WGS reads at the deleted *CRYL1* region. (**A**). IGV view of WGS reads from Patient #2 on the deleted region. The red box on the q12.11 chromosome 13 band indicates the position of the 272 kb window shown just below. Horizontal blue bars represent the whole *CRYL1* locus and part of the *IFT188* locus. Blue segments at the bottom correspond to the positions of OTOgenics^TM^ probes in the region. (**B**). Close-up of A at the centromeric (left) and telomeric (right) breakpoint regions. Black arrowheads mark the positions and orientations of the primers used in the first PCR designed to amplify the breakpoint region. Blue, green and red arrowheads represent the oligonucleotides used in the triple-primer PCR screening assay (as in Figure 1A).

**Figure 3 genes-16-00670-f003:**
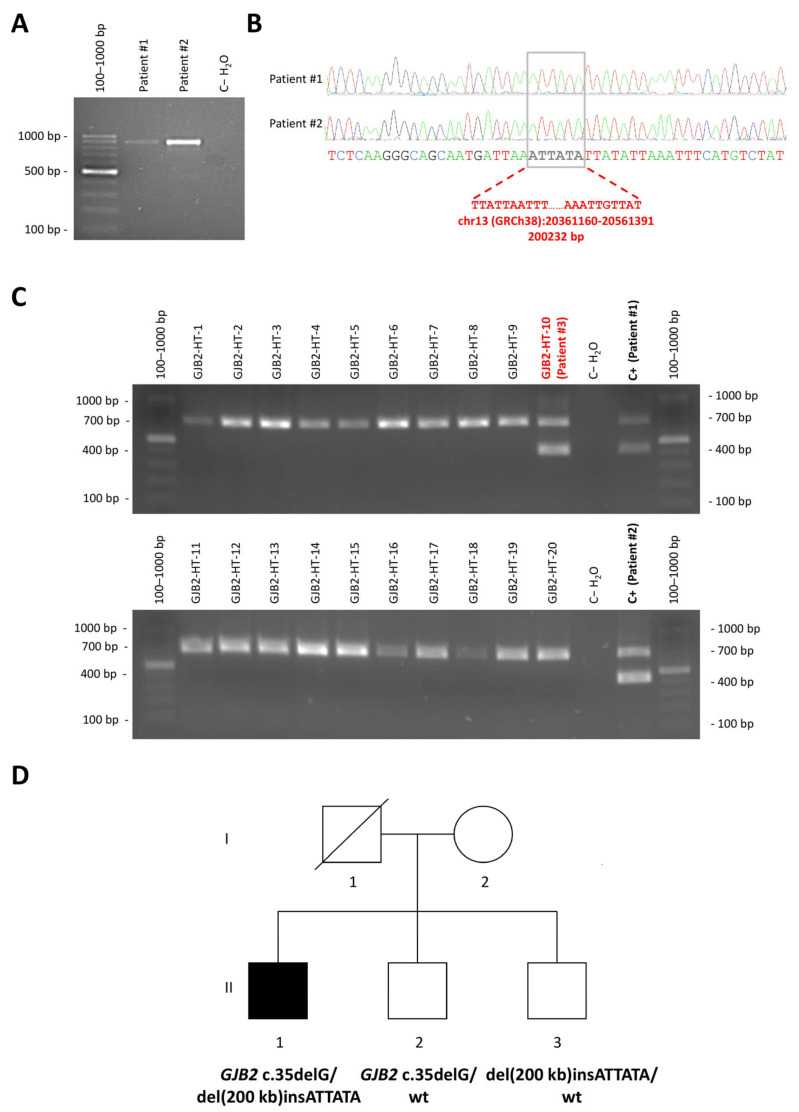
Characterization and screening of the novel 200 kb *CRYL1* deletion. (**A**). Agarose gel electrophoresis of products from first PCR designed to amplify the breakpoint region on germline DNA from Patient #1, Patient #2 and a control individual without the large 200 kb *CRYL1* deletion. An 897 bp band is observed only in the presence of the deletion. (**B**). Electropherograms corresponding to Sanger sequencing of the 897 bp bands obtained in *A* at the region of the breakpoints. The initial and final nucleotides of the deletion are shown in bold red characters. The inserted sequence (ATTATA) is shown in bold grey characters. (**C**). Agarose gel electrophoresis of products from the triple-primer PCR designed to screen for del(200 kb)insATTATA in 20 deaf patients previously evaluated in our laboratory, in whom *CRYL1* deletions had not been tested for, and who were heterozygous for a pathogenic/likely pathogenic *GJB2* variant. A 436 bp band is amplified from the allele affected by the deletion, whereas a 702 bp fragment is obtained from the wild-type allele. The DNA sample GJB2-HT16, corresponding to Patient #3 and labelled in bold red characters, contained 1 copy of del(200 kb)insATTATA. (**D**). Genogram of Patient #2. I and II indicate first and second generation, respectively. 1, 2 and 3 indicate first, second and third individual within the generation. Patient #2 is represented as II.1.

**Figure 4 genes-16-00670-f004:**
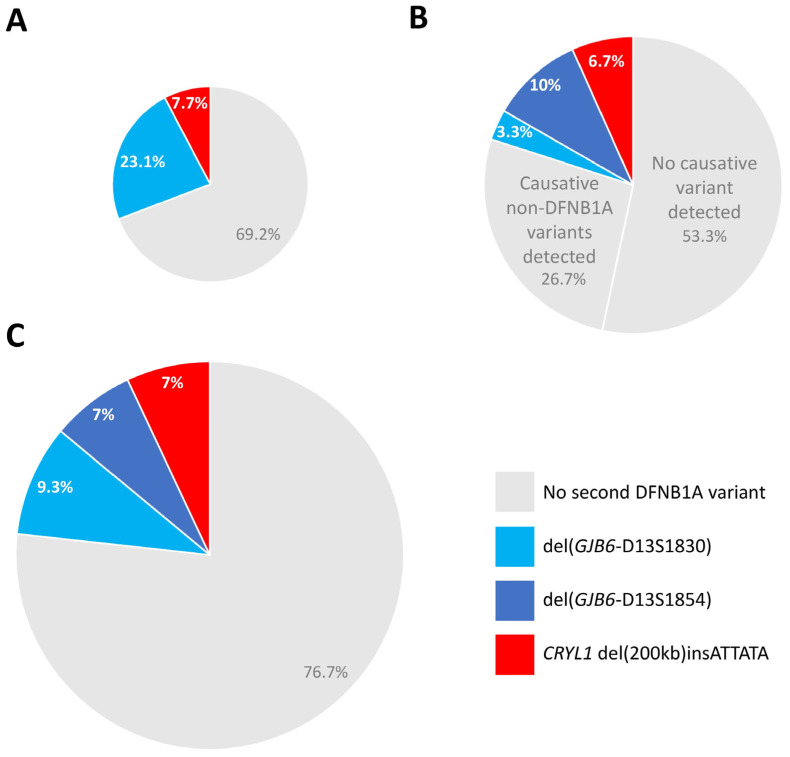
Relative contribution of the novel 200 kb *CRYL1* deletion to hearing loss in deaf patients with monoallelic pathogenic *GJB2* variants. Pie charts comparing the contribution of *CRYL1* del(200 kb)insATTATA (red), *GJB6*-D13S1854 (dark blue) and *GJB6*-D13S1830 (light blue) to the genetic diagnosis of deaf monoallelic carriers of pathogenic *GJB2* variants. Patients with no second DFNB1A variant detected are represented in grey. (**A**) Thirteen patients with no NGS panel testing requested. (**B**) Thirty patients with NGS panel testing requested (note the two distinct grey sectors corresponding to patients with no causative variant detected after panel testing and patients with causative non-DFNB1A variants detected). (**C**). Forty-three patients from panels A and B combined.

**Table 1 genes-16-00670-t001:** Large deletions not spanning the *GJB2* locus that cause hearing loss in compound heterozygosity with another pathogenic DFNB1A variant (see also Figure 1A).

Deletion	Genome Version	Centromeric Breakpoint	Telomeric Breakpoint	Reference
*GJB6*-D13S1830	GRCh38/hg38	chr13:20223038	chr13:20531828	[6]
	GRCh37/hg19	chr13:20797177	chr13:21105967	
*GJB6*-D13S1854	GRCh38/hg38	chr13:20228587	chr13:20460616	[7]
	GRCh37/hg19	chr13:20802726	chr13:21034755	
del(131 kb)	GRCh38/hg38	chr13:20365205	chr13:20496559	[8]
	GRCh37/hg19	chr13:20939344	chr13:21070698	
del(179 kb)	GRCh38/hg38	chr13:20347572	chr13:20526976	[9]
	GRCh37/hg19	chr13:20921711	chr13:21101115	
del(125kb)	GRCh38/hg38	chr13:20398369	chr13:20523824	[10]
	GRCh37/hg19	chr13:20972508	chr13:21097963	
**del(200 kb)insATTATA**	**GRCh38/hg38**	**chr13:20361160**	**chr13:20561391**	This study
	GRCh37/hg19	chr13:20935299	chr13:21135530	

## Data Availability

The data presented in this study are available on request from the corresponding author due to institutional limitations on publishing patient-derived whole genome sequencing results.

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
