# Peer review of "A Novel Recurrent 200 kb CRYL1 Deletion Underlies DFNB1A Hearing Loss in Patients from Northwestern Spain"

_genes, 2025, doi:10.3390/genes16060670_

Round 1
Reviewer 1 Report
Comments and Suggestions for Authors
The manuscript is well written, and the data is clear and easy to understand. The aim of the study was to identify the cause of hearing loss in patients. The patient was heterozygous for a known pathogenic mutation - GJB2 c.35delG. However, they examined copy number analysis of their NGS data which revealed reduced (50%) detection of 10 probes in the region near this gene indicating a deletion that had not been previously described (confirmed by sequencing). This deletion resulted in hearing loss when in compound heterozygosity with GJB2 c.35delG. They then designed a PCR based assay to test other deaf patients for this novel deletion. The manuscript describes the identification of a novel deletion in CRYL1 and in the region upstream of gene GJB2 in three unrelated deaf patients. The sibling of patient #2 also carries the deletion.
Line 325: should this read 3 out of 43 (7%) and not 7 out of 43?
Author Response
Reviewer #1's comments:
The manuscript is well written, and the data is clear and easy to understand. The aim of the study was to identify the cause of hearing loss in patients. The patient was heterozygous for a known pathogenic mutation - GJB2 c.35delG. However, they examined copy number analysis of their NGS data which revealed reduced (50%) detection of 10 probes in the region near this gene indicating a deletion that had not been previously described (confirmed by sequencing). This deletion resulted in hearing loss when in compound heterozygosity with GJB2 c.35delG. They then designed a PCR based assay to test other deaf patients for this novel deletion. The manuscript describes the identification of a novel deletion in CRYL1 and in the region upstream of gene GJB2 in three unrelated deaf patients. The sibling of patient #2 also carries the deletion.
Line 325: should this read 3 out of 43 (7%) and not 7 out of 43?
Response from the authors to Reviewer #1:
We thank reviewer 1 for his/her kind comments about our work. We are also grateful that he/she has pointed out the typo in Line 325, where, as he/she deduced, it should read "3 out of 43 (7%)" instead of "7 out of 43 (7%)". We have corrected this in the revised version of the manuscript (Line 348).
Reviewer 2 Report
Comments and Suggestions for Authors
This is an interesting paper identifying a novel 200kb deletion in patients with hearing loss and monoallelic GJB2 mutations.
- It is not clear to me from which data the authors conclude that the deletion and the GJB2 mutation for patients #1 and #3 are in trans. Only for patient #2 this is clear from the pedigree (Fig. 3D).
- Fig. 2 should be presented in a larger format (landscape) otherwise the details are not visible.
Author Response
Reviewer #2's comments:
This is an interesting paper identifying a novel 200kb deletion in patients with hearing loss and monoallelic GJB2 mutations.
- It is not clear to me from which data the authors conclude that the deletion and the GJB2 mutation for patients #1 and #3 are in trans. Only for patient #2 this is clear from the pedigree (Fig. 3D).
- Fig. 2 should be presented in a larger format (landscape) otherwise the details are not visible.
Response from the authors to Reviewer #2:
We thank the reviewer for his/her consideration of our work as interesting and for specifying improvable aspects of the manuscript:
Point #1:
In fact, we don't really conclude nor state in the manuscript that the large deletion and the GJB2 pathogenic variant are in trans in patients #1 and #3 . We only state that the large deletion is in trans with the c.35delG GJB2 pathogenic variant in Patient #2, as seen in Figure 3D. Unfortunately, we could not get access to DNA from parents/siblings of Patient #1 and Patient #3, so we could not confirm trans configuration in them. Given their phenotype, though, and the confirmed trans configuration in Patient #2, trans configuration is highly likely in both of them.
In case the reviewer was misled by the wording of sentences refering to Patient #1, we have edited the following sections:
Abstract: We have removed "in compound heterozygosity" from the sentence refering to "two unrelated deaf patients from Asturias" (Lines 26-28) (please, note that the abstract has been restructured into sections as per editorial request).
Results: We have reworded the sentence "... considering it as causative of DFNB1A in the patient in conjunction with GJB2 (NM_004004.6) c.35delG, as long as both variants affected different alleles (trans configuration)" to "... considering it as causative of DFNB1A in the patient in conjunction with GJB2 (NM_004004.6) c.35delG, provided that the variants affected different alleles (trans configuration)." (Lines 203-205).
Point #2:
In the revised version of the manuscript, we have enlarged Figure 2, enhanced its quality and changed its orientation to landscape to improve visibility. During the initial submission, the figure was provided as a separate file too, so the editorial office should be able to insert a higher-quality landscape image if the manuscript is accepted. We have also increased the quality of the remaining figures embedded within the manuscript.